# Involvement of IL-1β-Mediated Necroptosis in Neurodevelopment Impairment after Neonatal Sepsis in Rats

**DOI:** 10.3390/ijms241914693

**Published:** 2023-09-28

**Authors:** Zhimin Liao, Qing Zhu, Han Huang

**Affiliations:** Department of Anesthesiology and Key Laboratory of Birth Defects and Related Diseases of Women and Children, West China Second University Hospital of Sichuan University, Chengdu 610041, China; zhiminliao1982@163.com (Z.L.); zq82923@hotmail.com (Q.Z.)

**Keywords:** neonatal sepsis, neurodevelopment, IL-1β, necroptosis, rat

## Abstract

The mechanism of long-term cognitive impairment after neonatal sepsis remains poorly understood, although long-lasting neuroinflammation has been considered the primary contributor. Necroptosis is actively involved in the inflammatory process, and in this study, we aimed to determine whether neonatal sepsis-induced long-term cognitive impairment was associated with activation of necroptosis. Rat pups on postnatal day 3 (P3) received intraperitoneal injections of lipopolysaccharide (LPS, 1 mg/kg) to induce neonatal sepsis. Intracerebroventricular injection of IL-1β-siRNA and necrostatin-1 (NEC1) were performed to block the production of IL-1β and activation of necroptosis in the brain, respectively. The Morris water maze task and fear conditioning test were performed on P28–P32 and P34–P35, respectively. Enzyme-linked immunosorbent assay (ELISA), quantitative real-time PCR (RT-PCR), and Western blotting were used to examine the expression levels of proinflammatory cytokines and necroptosis-associated proteins, such as receptor-interacting protein 1 (RIP1) and receptor-interacting protein 3 (RIP3). Sustained elevation of IL-1β level was observed in the brain after initial neonatal sepsis, which would last for at least 32 days. Sustained necroptosis activation was also observed in the brain. Knockdown of IL-1β expression in the brain alleviated necroptosis and improved long-term cognitive function. Direct inhibition of necroptosis also improved neurodevelopment and cognitive performance. This research indicated that sustained activation of necroptosis via IL-1β contributed to long-term cognitive dysfunction after neonatal sepsis.

## 1. Introduction

It is well recognized that sepsis in early postnatal life is associated with neurodevelopment impairment [1,2,3,4]. Human studies have shown that 12–34% of neonatal sepsis survivors would have neurodevelopmental disorders [5,6,7]. In addition, neonatal sepsis is also considered as an independent risk factor for cerebral palsy [8,9] and psychiatric disorders, such as schizophrenia or autism, in later adult life [10,11,12,13,14].

As a newborn’s brain is vulnerable to various injuries, it is generally believed that neuroinflammation secondary to the systematic inflammation from neonatal sepsis is the major trigger event for late-onset neurodevelopment impairment [2,4,13,14]. Previous studies have demonstrated that there would be an increase in proinflammatory cytokine levels in the central nervous system (CNS) during sepsis [15,16,17,18,19]. Once inside the CNS, proinflammatory cytokines “inflame” the brain via multiple signaling pathways, especially by reactive gliosis. The activated microglia would undergo polarization into the M1 (pro-inflammatory) phenotype, resulting in further cytokine release. Our previous study showed that neonatal LPS injection resulted in sustained production of IL-1β, but not TNF-α or IL-6, in the brain [20]. It is well known that IL-1β is critical for maintaining brain hemostasis [21]. At the physiological level, IL-1 is widely involved in controlling the quantity of neural precursor cells and mature neurons [22,23], modulating synaptic refinement [24,25], and regulating the production of trophic factors, such as brain-derived neurotrophic factor (BDNF), nerve growth factor, and glutathione [26,27]. Physiological levels of IL-1β are also necessary for normal neurological function, such as hippocampal memory formation and sleep regulation [28]. However, excessive IL-1β is associated with cognitive deficits [19,29,30]. In line with early studies, our previous study also showed that blocking brain IL-1β production improved neurodevelopment following neonatal LPS exposure [20]. However, it remains unclear how neuroinflammation characterized by sustained IL-1β elevation leads to late-onset cognitive dysfunction.

Necroptosis is one type of programmed cell death, which is often activated in an inflammatory environment. Inflammatory stimuli lead to activation of intracellular adapter molecules (such as Fas-associated death domain (FADD) and TNF-receptor-associated death domain (TRADD)), which recruit receptor-interacting protein kinase 1 (RIP1), receptor-interacting protein kinase 3 (RIP3), and mixed lineage kinase domain-like protein (MLKL) to assemble the supramolecular necroptosis-inducing complex named the necrosome [31]. Finally, the necrosome causes cell membrane rupture and cell injury or damage [32,33]. Previous studies have shown that necroptosis is involved in various inflammation-related diseases, such as ischemia-reperfusion injury [34,35,36], traumatic or neurodegenerative diseases of the central nervous system [37,38], immune rejection after organ transplantation [39,40], and neonatal hypoxic/ischemic encephalopathy [36,41]. Our recent data also showed that activation of neuronal necroptosis was associated with neonatal LPS-exposure-induced neurodevelopment impairment [42].

Herein, we hypothesized that sustained elevated IL-1β activated necroptosis to cause neuron loss, which finally led to long-term cognitive dysfunction in a neonatal rat sepsis model.

## 2. Results

### 2.1. LPS Led to Impaired Neurodevelopment and Long-Lasting Activation of IL-1β in Brain

The experimental flow chart is shown in Experiment 1 of Figure 1. LPS injection led to 30% mortality in rat pups (Figure 2A) and a slower body weight increase in surviving pups during the first 2 weeks (*p* < 0.05, Figure 2B). From P28 to P31, rats were trained for the Morris water maze (MWM) test; then, rats were tested on P32. Traces of rats’ movement during the MWM test are represented in Figure 2C. There was no difference in the swimming speed between rats treated with LPS and normal saline (Figure 2D, *p* = 0.49), suggesting that LPS treatment did not cause locomotor impairment. Both the time spent in the target quadrant and the number of crossing quadrants were lower in LPS-treated rats (Figure 2E,F, both with *p* < 0.05). LPS-treated rats spent longer reaching the platform area (Figure 2G, *p* < 0.05). After training on P34, rats were tested on P35 for the fear-conditioning test. The freezing time was shorter in both the context test (Figure 2H, *p* < 0.001) and the cue tone test (Figure 2I, *p* < 0.01) in LPS-treated rats, suggesting impaired learning and memory. The contextual freezing time was closely associated with the expression level of IL-1β in the brain (*p* < 0.01 Figure 2J).

As expected, LPS elicited systematic inflammation, which resolved within 24 h (Figure 3A–C). Neuroinflammation was observed in the hippocampus as early as 8 h following LPS injection, and elevation of hippocampal IL-1β levels lasted for at least 32 days after LPS injection (Figure 3D–F).

### 2.2. IL-1β Served as a Critical Cytokine Promoting Impaired Neurodevelopment after Neonatal LPS Exposure

To evaluate whether sustained elevated IL-1β in the brain contributed to the late onset of neurodevelopment impairment, IL-1β expression was blocked with an intracerebroventricular IL-1β-siRNA injection (Experiment 2 in Figure 1). IL-1β-siRNA decreased the IL-1β mRNA level in the hippocampus after intraperitoneal LPS injection (Figure 4A, *p* < 0.05). As expected, intracerebroventricular injection of IL-1β-siRNA produced little effect on rat pups’ survival rate following LPS injection (Figure 4B, *p* = 0.129). The swimming speed was not different between groups, which suggested that *i.c.v.* injection of siRNA did not cause locomotor impairment either (Figure 4C, *p* = 0.281). The trace of the rats’ movement during the MWM test is represented in Figure 4D. IL-1β-siRNA treatment significantly increased the time spent in the target quadrant (Figure 4E, *p* < 0.05) and the number crossing the target quadrant (Figure 4F, *p* < 0.05), compared with rats receiving LPS+Control-siRNA. IL-1β-siRNA also shortened the time needed to reach the area for the platform, compared with LPS+Control-siRNA (Figure 4G, *p* < 0.05). Then, rats were trained for a fear-conditioning test on P34. The freezing time was longer in rats treated with IL-1β-siRNA in both the context test (Figure 4H, *p* < 0.01) and the cue tone test (Figure 4I, *p* < 0.05), compared with rats in the control-siRNA group.

Finally, hippocampal expression of essential proteins for neuronal development (GAP-43 and NeuN) and functional synapse (PSD-95 and SYN) were assessed (Figure 5A). IL-1β-siRNA treatment reversed LPS-induced low expression of GAP-43 (Figure 5B, *p* < 0.01) and NeuN (Figure 5C, *p* < 0.01). Although IL-1β-siRNA failed to improve the expression of PSD-95 (Figure 5D, *p* = 0.204) in LPS-treated rats, it improved the expression of SYN (Figure 5E, *p* < 0.05). In summary, blocking the expression of IL-1β in the brain could provide functional and molecular improvement in neurodevelopment impairment caused by neonatal LPS injection.

### 2.3. Necroptosis as the Potential Downstream Executer following Sustained IL-1β Activation

Next, we asked how increased IL-1β leads to impaired cognitive function. As indicated in Figure 5A, sustained IL-1β activation caused decreased expression of NeuN, suggesting a reduced quantity of viable neurons. Therefore, it is reasonable for us to hypothesize that elevated IL-1β level, as a marker of active inflammation, leads to neuronal damage or death and loss of neurons. Consistent with our previous report, neonatal LPS leads to activation of necroptosis on P35 but not apoptosis (Figure 6A–D). However, levels of MLKL were similar between groups (Figure 6E). With siRNA to inhibit expression of IL-1β, activation of necroptosis was also blocked. Then, the association between activation of necroptosis and impaired cognitive function was explored. The contextual freezing time was closely associated with the expression level of RIP1 and RIP3, as shown in Figure 6F,G. Finally, the causal relationship between IL-1β and necroptosis activation was investigated by observing the trajectory of expressions of IL-1β and necroptosis markers following LPS injection. As shown in Figure 6H,I, elevated IL-1β expression within the brain was observed as early as 2 h after LPS injection. However, an increase in RIP1 and MLKL was not observed until 4 h after LPS exposure. In summary, our data suggested that IL-1β-mediated necroptosis contributed to neuronal loss after neonatal LPS injection.

### 2.4. Impaired Neurodevelopment Caused by Neonatal LPS Injection Could Be Improved by Inhibiting Necroptosis

To determine the role of necroptosis in neonatal LPS injection-induced neurodevelopment deficit, necrostatin-1 (NEC1), a RIP-1-targeted necroptosis inhibitor, was injected into bilateral ventricles after *i.p.* LPS injection on P3 (Experiment 3 in Figure 1). As expected, the *i.c.v* injection of NEC1 failed to improve the survival rates of rat pups receiving the LPS injection (Figure 7A). After MWM training on P28-31, the MWM test was performed on P32. The swimming speed was not different between groups, which suggested that NEC1 did not cause locomotor impairment either (Figure 7B, *p* = 0.42). The trace of the rats’ movement during the MWM test is represented in Figure 7C. NEC1 treatment significantly increased the time spent in the target quadrant (Figure 7D, *p* < 0.01) and the number crossing the target quadrant (Figure 7E, *p* < 0.05), compared with treatment of LPS + dimethyl sulfoxide (DMSO) vehicle. NEC1 also shortened the time needed to reach the area for the platform, compared with LPS+DMSO (Figure 7F, *p* < 0.05). Then, rats were trained for a fear-conditioning test on P34. The freezing time was longer in rats treated with NEC1 in both the context test (Figure 7G, *p* < 0.001) and the cue tone test (Figure 7H, *p* < 0.001) compared with rats in the LPS+DMSO group.

Then, hippocampal expression of essential proteins for neuronal development (GAP-43 and NeuN) and functional synapse (PSD-95 and SYN) were assessed (Figure 8A). NEC1 treatment reversed LPS-induced low expression of GAP-43 (Figure 8B, *p* < 0.01) and NeuN (Figure 8C, *p* < 0.01). Also, NEC1 improved the expression of PSD-95 (Figure 8D, *p* < 0.01) and SYN (Figure 8E, *p* < 0.05). The inhibition efficacy of NEC1 on necroptosis was confirmed (Figure 8F). As expected, NEC1 treatment inhibited activation of RIP1 (Figure 8G, *p* < 0.05), RIP3 (Figure 8H, *p* < 0.05), and MLKL (Figure 8I, *p* < 0.01). However, it was surprising for us to find that NEC1 treatment also significantly reduced IL-1β mRNA expression levels (Figure 8J, *p* < 0.05).

## 3. Discussion

In the present study, we demonstrated that neonatal sepsis led to sustained neuroinflammation, characterized by elevated levels of IL-1β. Necroptosis was then activated to cause neuron death, resulting in molecular and behavioral abnormalities in neurodevelopment.

It has long been well-accepted that neuroinflammation plays a vital role in neurodevelopment impairment following neonatal sepsis [2,13,14]. After the systemic inflammation in sepsis survivors subsides, inflammation within the central nervous system will be sustained [17,18]. However, failure to identify the key cytokine(s) responsible for neonatal sepsis-induced neuroinflammation hindered the development of targeted therapeutics [43]. A previous observational study [19] has reported neonatal LPS exposure resulted in sustained elevation of interleukin-1β content and decreased neuron number in rat hippocampus when tested on day P71. Our study further demonstrated that by inhibiting IL-1β production in the brain, neurodevelopment could be improved, which indicated that IL-1β could be a potential therapeutic target against neurodevelopment impairment caused by neonatal sepsis [20].

Cell death is common following inflammation. However, neuron death via apoptosis is also a physiological process during neurodevelopment, which is necessary for the formation of functional synapses/innervation and clearance of excessive unconnected neurons [44]. Therefore, other forms of inflammation-related cell death may contribute to neuronal loss following neonatal sepsis. There was a transient cytokine storm within the CNS after a systematic inflammatory reaction during initial neonatal sepsis, which often leads to the activation of necroptosis. Ping Wang et al.’s study demonstrated that RIP1-dependent necroptosis activation, in part, was responsible for the systemic inflammatory response and organ injury in neonatal sepsis [45]. Dezhi Mu et al.’s study further suggested that toll-like receptor 9 (TLR9)-induced necroptosis was actively involved in sepsis-associated encephalopathy [46]. As shown in this current and our previous study [42], long-lasting activation of necroptosis was observed following activation of neuroinflammation after neonatal LPS injection. In our current study, there was a close association between levels of necroptosis-related protein and cognitive function. It was further found that prevention of necroptosis activation either by IL-1β-siRNA or RIP1 antagonist could improve the neurodevelopment deficit caused by neonatal LPS injection, suggesting that necroptosis contributed to neuronal loss or injury following neonatal sepsis. However, it was unexpected for us to find that RIP1 inhibition by NEC1 also reduced IL-1β levels, which indicates that there could be a bi-directional signal transduction between IL-1β and necroptosis. Furthermore, the potential reciprocal activation between IL-1β and RIP1 could be the leading cause of sustained neuroinflammation after neonatal sepsis, which warrants further investigation.

In addition to neuron quantity (reflected as NeuN level), the structure and function of synapses are of equal importance for normal cognition function. Synapsin (SYN) and postsynaptic density protein-95 (PSD-95) are key proteins in synaptic plasticity [47,48]. The SYN, located in the presynaptic membrane, is well characterized in modulating the release of neurotransmitters via fusion, endocytosis, and trans-membrane transporting [49]. PSD-95, located in the postsynaptic membrane, is involved in the regulation of the number and size of dendritic spines and the formation of glutamatergic synapses [50]. Levels of those proteins could serve as molecular markers for functional synaptic formation, while decreased expression indicates synaptic defects [51]. Nerve growth-associated protein (GAP-43) is closely related to neural development and axon regeneration and is well-accepted as a marker of nerve regeneration [52,53]. Our results showed that expression levels of these important proteins were decreased in survivors of septic neonatal rats, which was consistent with poorer test results in both the MWM and the fear-conditioning assessment. With the IL-1β knockdown or the RIP1 antagonist, the levels of these markers could be restored, and animal cognitive functions were also improved.

There are several limitations in this present study. It is interesting and of important clinical significance to determine the therapeutic window for IL-1β blockade, as it is not always possible for us to inhibit IL-1β at an early stage of sepsis. Furthermore, electrophysiological features of synaptic connection were not tested, but instead, key markers for functional synaptic connection were assessed, which was well recognized as a surrogate for synaptic function [47,50].

In conclusion, long-lasting neuroinflammation following neonatal sepsis was characterized by sustained IL-1β production within the brain, which subsequently led to necroptosis activation and neurodevelopment deficit. Cognitive function could be improved in rat pups who survived neonatal sepsis by inhibiting IL-1β production or necroptosis activation, indicating their potential role as therapeutic targets for cognitive disorders after early septic inflammation, as summarized in Figure 9. Future research should focus on elucidating the mechanism of bi-directional signal transduction between IL-1β and necroptosis and exploring the potential therapeutic window for neurodevelopment improvement by inhibiting IL-1β production in newborns with sepsis.

## 4. Materials and Methods

### 4.1. Experimental Animals

The experimental protocol was approved by the Committee for Experimental Animals of West China Second University Hospital (Approval Code: (2020) Animal Ethical No. (013); Approval Date: 27 March 2020). All the animals were treated in accordance with the Animal Research: Reporting of In Vivo Experiments (ARRIVE) guidelines. Sprague-Dawley rats on gestational day 19 were provided by the animal experiment center of Sichuan University and were kept in separate cages. After the offspring were delivered, they were kept with their mother with free access to food and water under a 12-hr light/dark cycle with constant temperature (21 ± 2 °C) and humidity (45–55%). Rat pups would be weaned from their mothers on post-natal day 21 (P21) and were housed in groups of five for each cage.

### 4.2. Reagents

Neonatal sepsis was induced in rat pups (both sexes) on P3 with intraperitoneal (*i.p.*) LPS (Sigma, St. Louis, USA, SMB00610) was dissolved in normal saline to 0.2 mg/mL and injected intraperitoneally at the dosage of 1 mg/kg. The pups were then returned to their dams. IL-1β-siRNA (sense sequence: 5′-GCA CAG ACC UGU CUU CCU ATT-3′; antisense sequence: 5′-UAG GAA GAC AGG UCU GUG CTT-3′, GenePharma, Shanghai, China, A03001) was prepared with RNAase-free-water and in vivo silencemag transfection reagent (OZ Biosciences, Marseille, France, SM30500) into a concentration of 0.08 nmol/μL. NEC1 (Abcam, Cambridge, UK, ab141053) was prepared with DMSO: ddH_2_O (1:19) into 1 mg/mL solution. IL-1β-siRNA and NEC1 were administrated with bilateral intracerebroventricular (*i.c.v.*) injection. In brief, the pups were placed on ice for 5 min to induce hypothermia anesthesia, which was confirmed by loss of response to gently paw squeezing as described in a previous study [54] and then mounted in a stereotaxic apparatus (RWD, Shenzhen, China). Then, 2 μL of reagent was injected at a rate of 100 nL/min (midpoint between the sagittal suture and the bregma, and ±1.5 mm to lateral, 1.5 mm for depth) [54]. The intracerebroventricular injection was performed 20 min after intraperitoneal injection.

### 4.3. Experimental Groups

Both sexes of rats were used in all the experiments. In experiment 1, animals in the LPS group received an *i.p.* injection of LPS as described before, while an *i.p.* injection of normal saline (NS) for animals in the Control group. In experiment 2, rats in the LPS+IL-1β-siRNA group received an *i.p.* injection of LPS and an *i.c.v.* injection of IL-1β-siRNA. In the LPS+control-siRNA group, rats received an *i.p.* injection of LPS and an *i.c.v.* injection of control-siRNA. In the Control group, rats received an *i.p.* injection of normal saline and an *i.c.v.* injection of control-siRNA. In experiment 3, rats in the LPS+NEC1 group received an *i.p.* injection of LPS and an *i.c.v.* injection of NEC1. In the LPS+DSMO group, rats received an *i.p.* injection of LPS and an *i.c.v.* injection of 5% DSMO in ddH_2_O (the solvent of NEC1). In the Control group, rats received an *i.p.* injection of normal saline and an *i.c.v.* injection of 5% DSMO in ddH_2_O.

For blood and brain samples, rats were sacrificed with an *i.p.* injection of 40 mg/kg sodium pentobarbital. Blood and hippocampus CA1 tissue samples were collected for subsequent tests such as quantitative real-time PCR (qRT-PCR), Enzyme-linked immunosorbent assay (ELISA), and Western blot.

### 4.4. Morris Water Maze Test

The Morris Water Maze (MWM) was used for spatial learning and memory function assessment [55]. The system (RWD Life Science, Shenzhen, China) consisted of a round poll (90 cm in diameter and 50 cm in depth), which was further divided into four quadrants, and a camera set 2 m above the poll to record the swimming trajectory of rats. Each quadrant of the poll was marked with an object on the wall of different colors and shapes, serving as clues for spatial vision. A round platform (10 cm in diameter) was placed in a fixed position 2 cm below the surface of black water and 30 cm away from the wall. The quadrant with the platform was defined as the target quadrant. During the experiment, the temperature of water was maintained at 30 ± 1 °C. Before testing, the rats were trained three times a day for 4 consecutive days (P28-P31). The rat was placed into the water with its face to the pool wall. If the rat found the platform within 90 s (sec) and stayed on the platform for 15 s, the time spent to find the platform was defined as the escape latency. Otherwise, the rat was guided to the platform to stay for 15 s, and the escape latency was recorded as 90 s. On testing day (P32), the platform was removed. The rat was put into the water from the quadrant opposite the target quadrant and was observed for 90 s. The time spent in the target quadrant and the number crossing the target quadrant were recorded. In the test session, escape latency was defined as the time spent to reach the area where the platform was placed during the training session. The swimming trajectory was analyzed by SMART software (Panlab, Barcelona, Spain).

### 4.5. Fear-Conditioning Test

The Fear-conditioning (FC) test was used to test both hippocampal-dependent and independent memory [56]. The FC system (ANY-maze, Stoelting, Wood Dale, IL, USA) consisted of a chamber with replaceable walls and a ceiling with an electrified grid floor. Training for fear-conditioning was conducted one day after the MWM test on P34. During the training session, the rat was first placed in the chamber and habituated to the surroundings for 2 min. Then, the rat received three pairs of conditional stimuli (a tone of 4000 Hz and 103 dB lasting for 20 s) and unconditional stimuli (an electrical shock of 0.5 mA to the feet lasting for 1 s and co-terminating with the tone). The context fear was tested on P35. Each rat was placed in a chamber identical to the training session but without cue tone or foot shock. After habitation for 2 min, the rat was observed for 3 min. The freezing behavior is defined as the complete absence of any physical movement except respiratory movement. As long as the rat was inside the chamber, its activity was continuously recorded by a camera mounted to the chamber ceiling. The video was later analyzed by ANY-maze video tracking system (Stoelting, Wood Dale, IL, USA) to determine the length of freezing time. One hour later, the cue-tone fear test was conducted. The rat would be placed in a novel chamber with different colors and ceiling shapes. After habituation for 2 min, a cue tone identical to the training session was played for 3 s. The test was repeated 3 times with an interval of 2 min, and the freezing time was also recorded.

### 4.6. Western Blot

The hippocampus samples were homogenized in cell lysis buffer (Cell Signaling, Danvers, MA, USA) with a sonicator. The samples were then centrifuged at 12,800 rpm. for 30 min at 4 °C, and the supernatant was collected. Total protein concentration was quantified using the Bradford protein assay (Beyotime Biotechnology, Shanghai, China). An equal amount of protein extracts was heated, denatured, and separated by electrophoresis. Then, the proteins were transferred to polyvinylidene difluoride membranes, which were blocked for 90 min at room temperature in 5% non-fat powdered milk in 0.1% Tween 20/tris-buffered saline (TBST) and were probed with the following primary antibodies. The primary antibodies included rabbit anti-RIP1 (1:1000, Cambridge, Abcam, UK, ab42126), rabbit anti-RIP3 (1:1000, Abcam, Cambridge, UK, ab62344), rabbit anti-MLKL (1:1000, Affinity, Houston, USA. DF7412), rabbit anti-Caspase-3 (1:1000, Cell signaling, Danvers, MA, USA, 9662S), rabbit anti-GAP-43 (1:500, Abcam, Cambridge, UK, ab16053), rabbit anti-NeuN (1:500, Abcam, Cambridge, UK, ab128886), rabbit anti-Synapsin (1:500, Abcam, Cambridge, UK, ab64581), rabbit anti-PSD-95 (1:500, Abcam, Cambridge, UK, ab18258), and rabbit anti-β-actin (1:100,000, ABclonal, Woburn, MA USA, AC026). Subsequently, the membranes were incubated with horseradish peroxidase-conjugated anti-rabbit (1:5000, Proteintech, Wuhan, China, B900210) for 1 h at room temperature. The protein bands were captured with the Chemidoc XRS system (Bio-Rad) after treatment with chemiluminescence reagents (ECL; Amersham Pharmacia Biotech, Piscataway, NJ, USA). We used Image-Pro Plus 5.0 (Media Cybernetics, Rockville, Maryland, USA) to perform grayscale analysis for each protein band. Results were normalized to levels of the housekeeping protein β-actin and were expressed as a ratio relative to the control for data analysis. Microsoft visio 2013 (Microsoft, Redmond, Washington, USA) was used to generate all the figures from the original images.

### 4.7. Quantitative Real-Time PCR

RNA extraction kits (#RE-03011 from Foregene, Chengdu, China) were used to extract the total mRNA of the hippocampus. According to the manufacturer’s instruction, the hippocampus was placed in 500 μL lysis buffer and then homogenized on ice. After being centrifuged at 12,000 rcf at 4 °C for 20 min, the supernatants were collected, and total RNA was extracted by removing protein, DNA, and salt ions. cDNA was generated from 1000 ng of total RNA using the PrimeScript RT reagent kit (#RR047A and RR047B from Takara Bio, Shiga, Japan) according to the manufacturer’s instructions. TB Green (#639676 from Takara Bio, Japan) was used to perform the RT-qPCR for detecting the expression of IL-1β mRNA in the hippocampus. There were 3 uL of cDNA, 10 μL of TB Green^®^ Advantage^®^ qPCR Premix (#639676 from Takara Bio, Shiga, Japan), and 0.8 μM of each primer in the final volume of 20 μL supplemented with ddH_2_O. qPCR reactions were set up in triplicate in a ThermoFisher thermocycler (ThermoFisher, Shanghai, China) as follows: initial denaturation for 30 s at 95 °C, followed by 45 cycles of 5 s at 95 °C, 30 s at 55 °C, and 30 s at 72 °C. The relative mRNA expression level of the target gene was calculated with the 2^−ΔΔCT^ method, with β-actin serving as the control. The primer sequences used are as follows:·β-actin forward:5′-GAAGATCAAGATCATTGCTCCT-3′;·β-actin reverse: 5′-TACTCCTGCTTGCTGATCCA-3′;·IL-1 β forward: 5′-ATCCTCCAGTCTCCTTGTG-3′;·IL-1 β reverse: 5′-AGCTCTTGTGTCGCTGTGA-3′.

### 4.8. Enzyme-Linked Immunosorbent Assay (ELISA)

ELISA was used to quantify the levels of proinflammatory cytokines, including TNF-α, IL-6, and IL-1β, in the blood serum and the hippocampus. Blood was centrifuged at 2000× *g* for 20 min, and the supernatants were collected and stored at −80 °C until the test. The hippocampal protein was extracted, as mentioned in Section 4.6. ELISA kits were purchased from Mlbio Biotechnology (Mlbio, Shanghai, China), and analysis was performed following the manufacturer’s instructions. Absorbance at 450 nm was measured with a Tecan Sunrise^TM^ microplate reader with correction at 680 nm wavelength. The protein concentration was determined by the measured optical density of the reaction according to the standard samples.

### 4.9. Statistical Analysis

SPSS19.0 (SPSS Inc., Chicago, IL, USA) was used for statistical analysis. The data were expressed as mean ± standard deviation. Comparison between the study groups was analyzed with one-way analysis of variance (ANOVA), followed by post hoc Student Newman–Keuls test if indicated. The correlation analysis was performed with Pearson regression, and *p* < 0.05 was considered statistically significant.

## Figures and Tables

**Figure 1 ijms-24-14693-f001:**
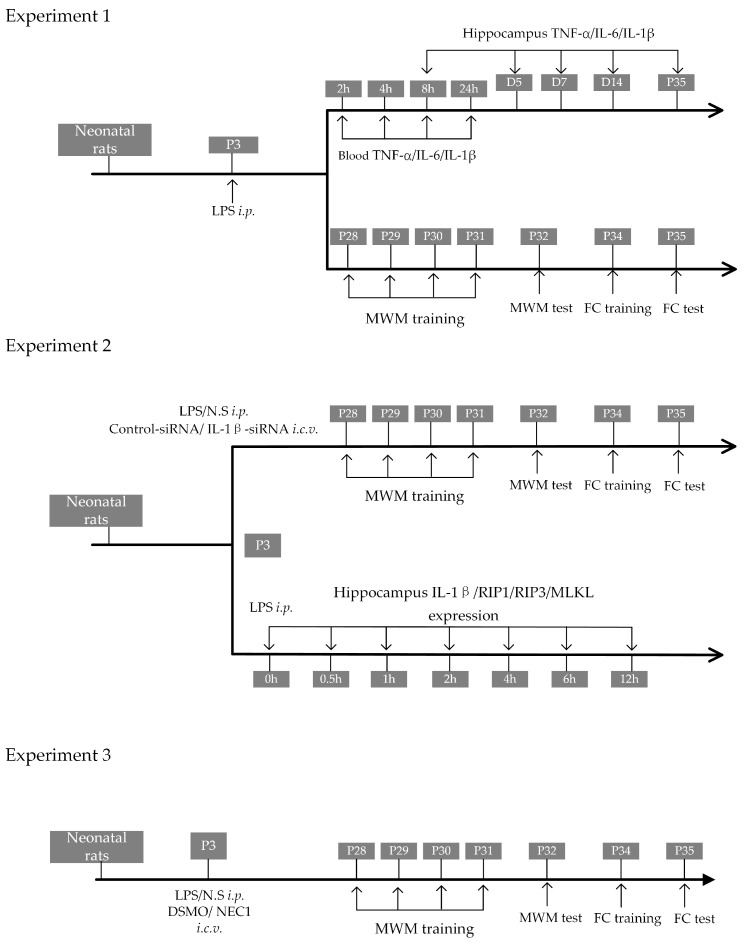
Flow chart illustrating the experimental designs. Experiment 1 is designed to test the association between neonatal sepsis and cognitive impairment. Experiment 2 is designed to investigate the role of IL-1β on neurodevelopment after neonatal sepsis. Experiment 3 is designed to investigate the role of necroptosis on neurodevelopment after neonatal sepsis. LPS: lipopolysaccharide; NS: normal saline; NEC1: necrostatin; DMSO: dimethyl sulfoxide; MWM: Morris water maze; FC: fear conditioning; *i.p.*: intraperitoneal injection; *i.c.v.*: intracerebroventricular injection.

**Figure 2 ijms-24-14693-f002:**
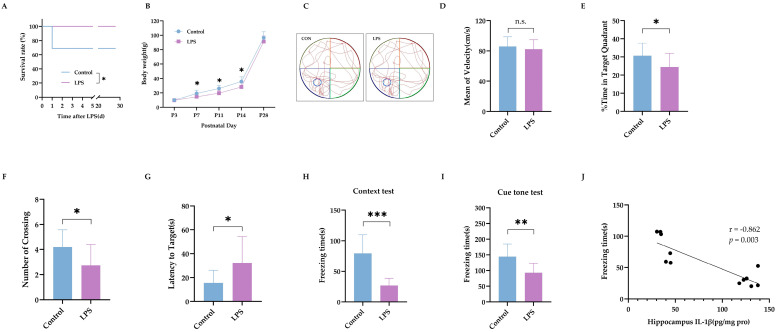
Intraperitoneal LPS injection on P3 led to long-term cognitive impairment. (**A**) The survival rates of rat pups following LPS injection (*n* = 16). (**B**) The increase in body weight in rat pups (*n* = 16 in the control group, *n* = 11 in the LPS group). (**C**) Representative traces of the MWM test. (**D**) The mean velocity of swimming during the MWM test. (**E**) Percentage of time spent in the target quadrant. (**F**) Number of platform crossings. (**G**) Latency time to find the area for the platform (*n* = 16 in the control group, *n* = 11 in the LPS group for (**D**–**G**)). (**H**) The freezing time of rats in the context test. (**I**) The freezing time of rats in the cue tone FC test (*n* = 16 in the control group, *n* = 11 in the LPS group for (**H**,**I**)). (**J**) The correlation between hippocampal IL-1β level and freezing time of context test (*n* = 6). LPS: lipopolysaccharide; MWM: Morris water maze; FC: fear conditioning. Data are expressed as the mean ± SD. * *p* < 0.05, ** *p* < 0.01, *** *p* < 0.001, n.s.: no significance.

**Figure 3 ijms-24-14693-f003:**
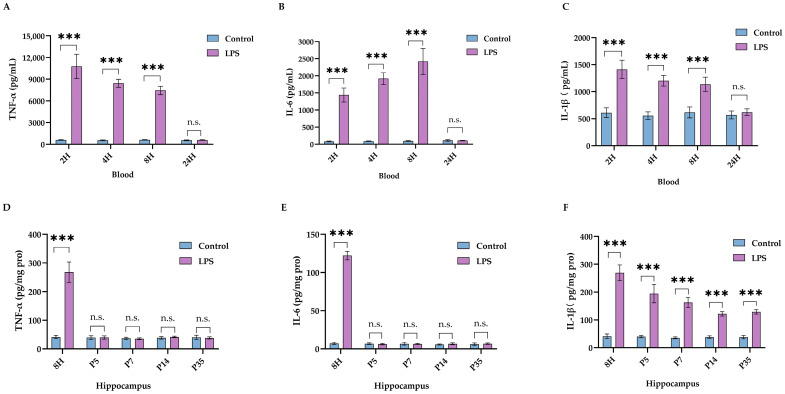
Intraperitoneal LPS injection on P3 led to long-lasting activation of IL-1β in the hippocampus. Changes in blood levels of TNF-α (**A**), IL-6 (**B**), and IL-1β (**C**) following LPS injection (*n* = 6 for each group at each time point). Changes in hippocampal content of TNF-α (**D**), IL-6 (**E**), and IL-1β (**F**) following LPS injection on P3 (*n* = 6 for each group at each time point). Data are expressed as the mean ± SD. *** *p* < 0.001, n.s.: no significance.

**Figure 4 ijms-24-14693-f004:**
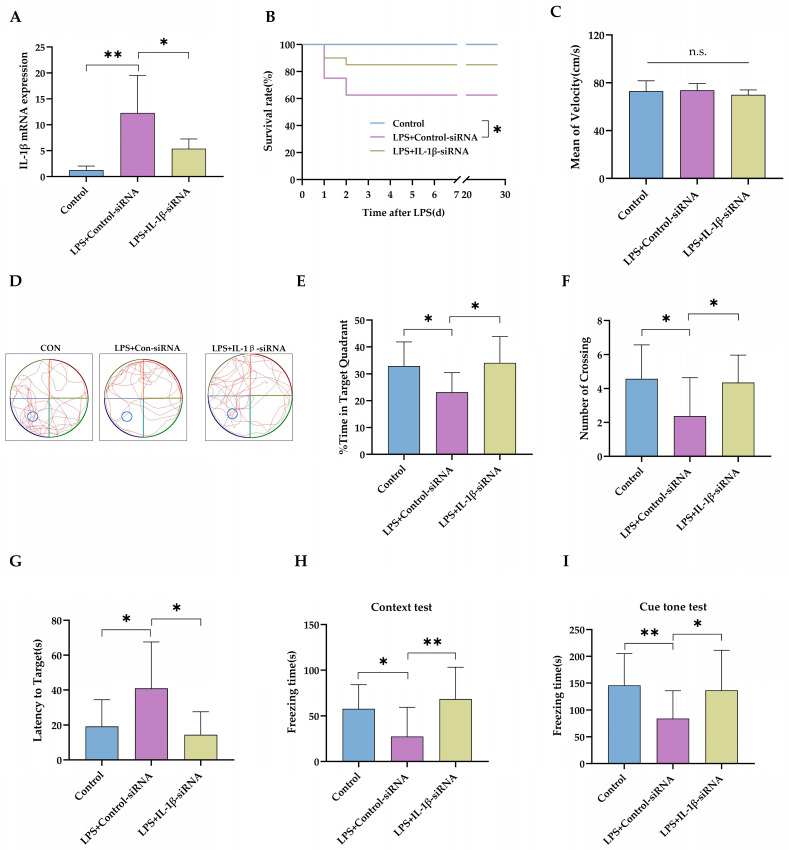
Knockdown of the IL-1β expression by siRNA improved the long-term cognitive function after intraperitoneal LPS injection. (**A**) PCR results show the knockdown efficiency of IL-1β-siRNA (*n* = 6). (**B**) The survival rates of rat pups following LPS injection (*n* = 14 in group Control, *n* = 16 in group LPS+Control-siRNA, *n* = 20 in group LPS+IL-1β-siRNA). (**C**) Mean swimming velocity during the MWM test (*n* = 14 in group Control, *n* = 10 in group LPS+Control-siRNA, *n* = 17 in group LPS+IL-1β-siRNA for (**C**,**E**–**G**)). (**D**) Representative traces of the MWM test. (**E**) Percentage of time spent in the target quadrant. (**F**) Number of platform crossings. (**G**) Latency time to find the area for the platform. (**H**) The freezing time of rats in the context of the FC test. (**I**)The freezing time of rats in the cue tone test (*n* = 14 in group Control, *n* = 10 in group LPS+Control-siRNA, *n* = 17 in group LPS+IL-1β-siRNA for (**H**,**I**)). LPS: lipopolysaccharide; MWM: Morris water maze; FC: fear conditioning. Data are expressed as the mean ± SD. * *p* < 0.05, ** *p* < 0.01, n.s.: no significance.

**Figure 5 ijms-24-14693-f005:**
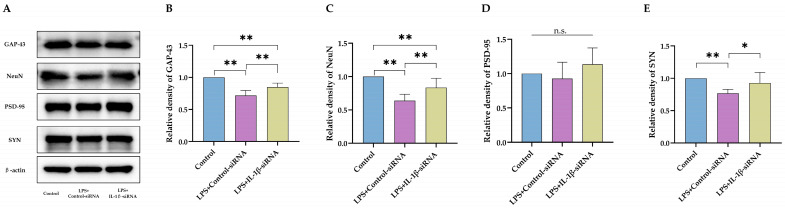
Knockdown of the IL-1β expression by siRNA reduced impairment of neuron development and synaptic function. (**A**) Western blotting with statistics for expression of GAP-43 (**B**), NeuN (**C**), PSD-95 (**D**), and SYN (**E**) (*n* = 6). LPS: lipopolysaccharide. Data are expressed as the mean ± SD. * *p* < 0.05, ** *p* < 0.01, n.s.: no significance.

**Figure 6 ijms-24-14693-f006:**
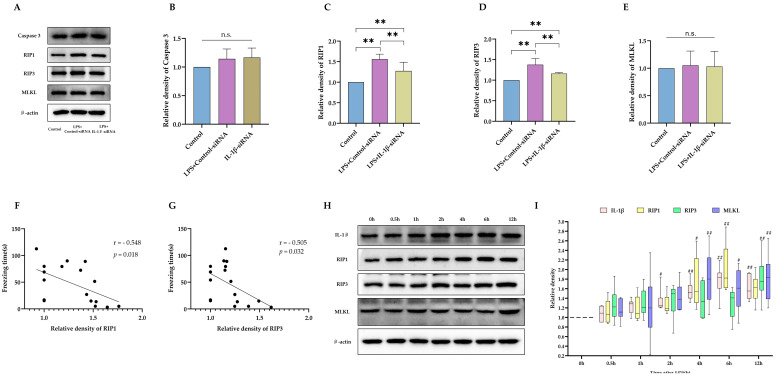
Necroptosis is the potential downstream executor following sustained IL-1β activation. (**A**) Western blotting results of Caspase3, RIP1, RIP3, and MLKL expression levels. Statistics in the expression of Caspase3 (**B**), RIP1 (**C**), RIP3 (**D**), and MLKL (**E**) protein in WB (*n* = 6). Correlation between context test freezing time and RIP1 (**F**) and RIP3 (**G**) (*n* = 6). (**H**) Time course of hippocampal expression levels of IL-1β and necroptosis-related proteins following neonatal LPS injection with statistics (**I**) (*n* = 6 for each protein at each time point). LPS: lipopolysaccharide; Data are expressed as the mean ± SD. ** *p* < 0.01,^#^ *p* < 0.05 vs. 0 h, ^##^ *p* < 0.01 vs. 0 h. n.s.: no significance.

**Figure 7 ijms-24-14693-f007:**
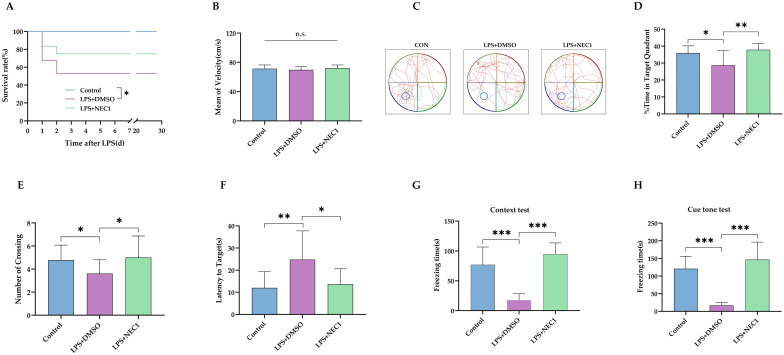
Inhibiting necroptosis improved the long-term cognitive function after neonatal LPS injection. (**A**) The survival rates of rat pups following LPS injection (*n* = 9 in group Control, *n* = 34 in group LPS+DMSO, *n* = 12 in group LPS+NEC1). (**B**) Mean velocity of swimming during the MWM test. (**C**) Representative traces of the MWM test. (**D**) Percentage of time spent in the target quadrant. (**E**) Number of platform crossings. (**F**) Latency time to find the area for the platform (*n* = 9 in group Control, *n* = 18 in group LPS+DMSO, *n* = 9 in group LPS+NEC1 for (**B**,**D**–**F**)). (**G**) The freezing time of rats in the context of the FC test. (**H**)The freezing time of rats in the cue tone test (*n* = 9 in group Control, *n* = 18 in group LPS+DMSO, *n* = 9 in group LPS+NEC1 for (**G**,**H**)). LPS: lipopolysaccharide; NEC1: Necrostatin-1; DMSO: dimethyl sulfoxide; MWM: Morris water maze; FC: fear conditioning. Data are expressed as the mean ± SD. * *p* < 0.05, ** *p* < 0.01, *** *p* <0.001, n.s.: no significance.

**Figure 8 ijms-24-14693-f008:**
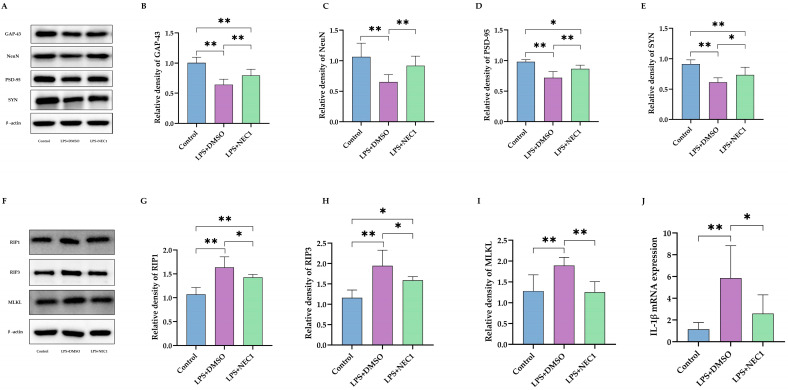
Inhibiting necroptosis reduced impairment of neuron development and synaptic function after neonatal LPS injection. (**A**) Western blotting with statistics for expression of GAP-43 (**B**), NeuN (**C**), PSD-95 (**D**), and SYN (**E**) (*n* = 6). (**F**) Western blotting with statistics for expression of RIP1 (**G**), RIP3 (**H**), and MLKL (**I**) (*n* = 6). (**J**) IL-1β mRNA expression in hippocampus after NEC1 treatment (*n* = 6). LPS: lipopolysaccharide; DMSO: dimethyl sulfoxide; NEC1: Necrostatin-1. Data are expressed as the mean ± SD. * *p* < 0.05, ** *p* < 0.01, n.s.: no significance.

**Figure 9 ijms-24-14693-f009:**
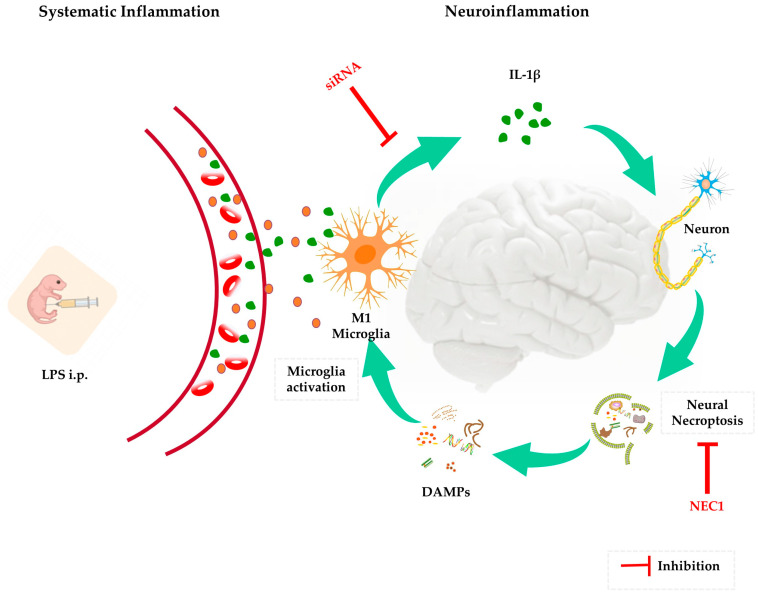
A schematic summarizing the mechanism and conclusion.

## Data Availability

The data presented in this study are available upon reasonable request from the corresponding author.

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
