# Peer review of "Involvement of IL-1β-Mediated Necroptosis in Neurodevelopment Impairment after Neonatal Sepsis in Rats"

_ijms, 2023, doi:10.3390/ijms241914693_

Round 1

Reviewer 1 Report

In this work, the authors determined whether neonatal sepsis-induced long-term cognitive impairment was associated with activation of necroptosis. Rat pup on postnatal day 3 (P3) received intraperitoneal injection of lipopolysaccharide 13 (LPS, 1 mg/kg) to induce neonatal sepsis.

The idea of this study - is interesting; nevertheless, this manuscript needs some improvements and corrections before publishing may be possible.

General points:

Please add a list of abbreviations before References section to your manuscript.

Special points:

Please add as Figure 1 the time-line of all your experiments to your manuscript and please delete the part A from the Figures 1, 2, 4.

All Figures are too small, please make all Figures bigger. You can add more separately Figures.

Introduction

The Introduction section is too short, please describe all studies done by you and exists on this topic up to date.   

Lines 28-29: please add multiple references at the end of this sentence.

Lines 33-42: please add multiple references at the end of each these sentences.

Discussion

Lines 191-200: please add multiple references at the end of each these sentences.

Line 196: please describe all these studies exactly.

Lines 206: please describe all these studies exactly.

Please add the Future perspectives section to your manuscript.

Materials and Methods

First of all, please add to this section the exactly name of the organization, the exactly date and the number of the permission of all your experiments.

Please add a total number of the animals and the sex of the animals used in all your experiments.

Please add the exactly number and the sex of the animals used for each method.

Lines 256-267: please describe the injection more exactly.

Line 263: please describe the anesthesia of the pups very exactly.

Please add to each method the appropriate references according to which group or publication you did this method.

Lines 284-286: please add more exactly product information about this system used by you.

Lines 303-306: please add more exactly product information about this chamber used by you.

Conclusion

Please add more Conclusion and Future Perspectives section to your manuscript.

Author Response

Response to the reviewers’ comments

We appreciate the constructive suggestions from the reviewers and the editor. We have revised our manuscript according to your advice. In the revised manuscript, the corrections are marked in red.

Reviewer #1:

Q1: Please add a list of abbreviations before References section to your manuscript.

Reply: We have added the abbreviations list at the end of the manuscript. Please refer to Line 364-385.

Q2: Please add as Figure 1 the time-line of all your experiments to your manuscript and please delete the part A from the Figures 1, 2, 4. All Figures are too small, please make all Figures bigger. You can add more separately Figures.

Reply:Thanks for your insightful advices! We have re-arranged the flowcharts into one single figure as Figure 1. Rest figures were also enlarged and divided into several more figures.

Q3: The Introduction section is too short, please describe all studies done by you and exists on this topic up to date.  Lines 28-29: please add multiple references at the end of this sentence.

Reply: We extended the introduction section and more related references were added. Our previous studies were explained in Line 44-45 and Line 55-56, respectively.

Q4: Lines 33-42: please add multiple references at the end of each these sentences.

Reply: We added more related references, please refer to reference 13-30 in the revised manuscript.

Q5: Lines 191-200: please add multiple references at the end of each these sentences.

Reply: We added more related references to each of these sentences, please refer to Line 189-196.

Q6: Line 196: please describe all these studies exactly.

Reply: Detailed description was added in Line 193-194.

Q7: Lines 206: please describe all these studies exactly.

Reply: Detailed description was added in Line 201-204.

Q8: Conclusion

Please add the Future perspectives section to your manuscript.

Reply: Thanks for your insightful advice. Future perspectives were added in Line 233-235.

Q9: Materials and Methods

First of all, please add to this section the exactly name of the organization, the exactly date and the number of the permission of all your experiments.

Reply: The detailed approval information was added as “The experimental protocol was approved by the Committee for Experimental Animals of West China Second University Hospital (Approval Code: (2020) Animal Ethical No. (013); Approval Date: 2020.03.27).” Please refer to Line 239-420.

Q10: Please add a total number of the animals and the sex of the animals used in all your experiments.

Please add the exactly number and the sex of the animals used for each method.

Reply: we cannot agree with you any more on this. Both sexes of neonatal rats were used in this study, which was stated in Line 247 and 259. The exact number of animals used in each experiment group was stated in each corresponding figure legend.

Q11: Lines 256-267: please describe the injection more exactly.

Line 263: please describe the anesthesia of the pups very exactly.

Reply: Details of LPS injection was added in Line 247-248. Hypothermia anesthesia was described in Line 253-254.

Q12: Lines 284-286: please add more exactly product information about this system used by you.

Reply: The production information was added in Line 271-273. Thanks!

Q13: Lines 303-306: please add more exactly product information about this chamber used by you.

Reply: The production information of the chamber was added in Line 287-288. Thanks!

Reviewer 2 Report

Liao et al. "Involvement of IL-1β-mediated Necroptosis in Neurodevelopment Impairment After Neonatal Sepsis in Rats" is an interesting study in which the authors found sustained elevation of IL-1β level after initial neonatal sepsis first and subsequent activation of necroptosis. Further knockdown of IL-1B expression had a beneficial effect on cognitive function which is linked with inhibition of necroptosis.
The strength of the article is that it investigates the sustained IL-1B productions following neonatal sepsis resulting in necroptosis activation and neurodevelopmental deficits. This result was further validated using the Knockdown approach, behavioral approach, and exploiting the molecular biological tools and techniques.

However, there are weaknesses in the article, which should be properly addressed below.

1)Introduction section:
a. Line 37-40: Your recent data could better fit into the last part of the introduction section. So you can move it to there. Assuming other labs are also working on similar experiments, please introduce relevant explanations.

b.The introduction section lacks lacking broad literature review and generalized concepts.

 For example, the introduction section should elaborate on the contribution of all neural cell types including glia with a focus on IL-B in neurological diseases. For more details, please see
 PMID: 36768596. Introducing a few sentences or paragraphs between lines 38-46 will be meaningful.

c. Please explain homeostasis in the brain as well as how sustained activation through different downstream signaling cascades may lead to the development of neuroinflammation and diseases. For the homeostasis concept, you may introduce an antioxidant molecule like glutathione. See PMID: 35011559.

Line 69-70: In Figure 1 (A to Q): Make sure that text size, style, and fonts in all the figures throughout the manuscript are consistent. I see some of the text on the x-axis is stretched (Figure 1, O) and the spelling is not correct for "control" (Figure 1, K).

Figure 2 and the rest of the figures: See the comments for Figure 1, if any discrepancies are seen. The text in the figure should be visible and printable clearly on the paper.

The text size for '*' can be increased here and elsewhere.

- Make sure if any abbreviation is used for the first time, in the manuscript need to provide the full form.

 2) Method section:
Line 320: "lysisbuffer" Please check if those types of spacing errors are found in the manuscript elsewhere as well.
Line 330: space error "1:1000Cell signaling"

Line 331-332: Provide the dilution used if available for the anti-PSD-95
- Any secondary antibodies used or were these primary antibodies already tagged?
Line 335-345: Please briefly explain what reagents those kits contain and how they help in isolation.
Line 336: "hippocampal nuclear"- check this wording is correct.

-Line 332-333: Please specify what software you used after you obtained the images to write text or process images.

-It would be useful to provide the Catalogue number and lot number if possible for the antibody or relevant information for the reagents if available.
- Please briefly describe, how you performed cDNA synthesis and real-time PCR as well as ELISA. The main brief points will be useful. Readers may find it easier to get clearer pictures already here from your manuscript rather than going into detail about the instructions manual from the vendor.

Original images:
When I see the original blot, the blot looks clearer and the figures have more clarity. For some blots, I could not see the standard markers. When you compiled the final figure, that original blot appeared a little hazy or stretched. I recommend sticking to the original blot and the compiled blot should look more towards the original signal. This applies to all of the figures.
 In addition, when I look at the original blot for the file name Figure 3H RIP3.TIF, I see several different signal traces, and I cannot understand which signal traces belong to RIP3. Maybe it would be better to specify somewhere.

Supplemental data:
-While clicking on the link in the supplemental section, I could not find any data. Please provide the data so that I can review it.

If you can summarize your findings in the schematics form including the unknown or future direction, and provide those figures for the underlying mechanism supporting your hypothesis, that would be a great addition. These schematics can be introduced in the discussion section.

Overall, the original article uses a different multidisciplinary approach to validate the hypothesis, and the presentation style and another issue I have raised must be addressed properly. I would need to review all the supplemental figures before I endorse it.

Author Response

Response to the reviewers’ comments

We appreciate the constructive suggestions from the reviewers and the editor. We have revised our manuscript according to your advice. In the revised manuscript, the corrections are marked in red.

Reviewer #2:

Q1: 1)Introduction section:

  1. Line 37-40: Your recent data could better fit into the last part of the introduction section. So you can move it to there. Assuming other labs are also working on similar experiments, please introduce relevant explanations.

Reply: Thanks for your kind suggest. We re-organized this part and please refer to Line 32-56. Your advice is very helpful!

Q2: b.The introduction section lacks lacking broad literature review and generalized concepts.
 For example, the introduction section should elaborate on the contribution of all neural cell types including glia with a focus on IL-B in neurological diseases. For more details, please see PMID: 36768596. Introducing a few sentences or paragraphs between lines 38-46 will be meaningful.

Reply: Thanks for your kind suggest. The role of IL-1β was described in Line 38-44. The paper you mentioned is cited as reference no. 27. Thanks!

Q3: Please explain homeostasis in the brain as well as how sustained activation through different downstream signaling cascades may lead to the development of neuroinflammation and diseases. For the homeostasis concept, you may introduce an antioxidant molecule like glutathione. See PMID: 35011559.

Reply: Please refer to our re-organized introduction from Line 35-44, which was revised by integrating your advice. The paper you mentioned is cited as reference no. 26. Thanks!

Q4: Line 69-70: In Figure 1 (A to Q): Make sure that text size, style, and fonts in all the figures throughout the manuscript are consistent. I see some of the text on the x-axis is stretched (Figure 1, O) and the spelling is not correct for "control" (Figure 1, K).

Reply: Apologized for these confusions. We re-arranged all the figures and check the text and spelling thoroughly! Thanks!

Q5: Figure 2 and the rest of the figures: See the comments for Figure 1, if any discrepancies are seen. The text in the figure should be visible and printable clearly on the paper. The text size for '*' can be increased here and elsewhere.

Reply: Apologized for these confusions. We re-arranged all the figures! Thanks!

Q6:  Make sure if any abbreviation is used for the first time, in the manuscript need to provide the full form.

Reply: We added the full form of abbreviation at its first appearance. Thanks for this kind reminding.

Q7:  2) Method section:

Line 320: "lysisbuffer" Please check if those types of spacing errors are found in the manuscript elsewhere as well

Line 330: space error "1:1000Cell signaling"

Reply: These errors were corrected. Please refer to Line 302 and 309, respectively. Thanks!

Q8: Line 331-332: Provide the dilution used if available for the anti-PSD-95
- Any secondary antibodies used or were these primary antibodies already tagged?
Reply: We added the dilution of anti-PSD-95 at Line 311. The secondary antibody was used for detecting protein bands and was described in Line 312-313.

Q9: Line 332-333: Please specify what software you used after you obtained the images to write text or process images.

Reply: Microsoft visio 2013 (Microsoft, U.S) was used to generate all the figures from original images, as we stated in Line 317.

Q10: Line 335-345: Please briefly explain what reagents those kits contain and how they help in isolation. Line 336: "hippocampal nuclear"- check this wording is correct.

Reply: RNA isolation and extraction was described in Line 319-322. And the wording error is also corrected. Thanks.

Q10: -It would be useful to provide the Catalogue number and lot number if possible for the antibody or relevant information for the reagents if available.

Reply: We added all of the Catalogue number of antibody in Line 307-312.

Q11: Please briefly describe, how you performed cDNA synthesis and real-time PCR as well as ELISA. The main brief points will be useful. Readers may find it easier to get clearer pictures already here from your manuscript rather than going into detail about the instructions manual from the vendor.

Reply: The brief description of RT-PCR was added in Line 325-328.

Q12: When I see the original blot, the blot looks clearer and the figures have more clarity. For some blots, I could not see the standard markers. When you compiled the final figure, that original blot appeared a little hazy or stretched. I recommend sticking to the original blot and the compiled blot should look more towards the original signal. This applies to all of the figures.
In addition, when I look at the original blot for the file name Figure 3H RIP3.TIF, I see several different signal traces, and I cannot understand which signal traces belong to RIP3. Maybe it would be better to specify somewhere.

Reply: Thanks very much for this insightful advice. We rearrangement the figures and use the original blot to avoid stretch. We added a red arrow to point out the RIP3 traces in the original image file named Figure 6A RIP3.

Q13: Supplemental data:

While clicking on the link in the supplemental section, I could not find any data. Please provide the data so that I can review it.

Reply: As a matter of factor, there is no supplemental data in our current manuscript. Thanks for your concern.

Q14: If you can summarize your findings in the schematics form including the unknown or future direction, and provide those figures for the underlying mechanism supporting your hypothesis, that would be a great addition. These schematics can be introduced in the discussion section.

Reply: Thanks for your wonderful comment. A summarizing graph was added as figure 9.

Round 2

Reviewer 1 Report

Thank you for all corrections.

Author Response

We would like to thank this reviewer for the supportive and helpful comments.

Reviewer 2 Report

The article has been now improved now. However, there are still some improvements the authors need to make.

For example:

1) Line 72: "was showed": I am not checking the English language, please make as needed correction through out the manuscript. I can suggest to run your manuscript with native English speaker or Professionals.

2) Please consider consistency in the text within the figures to be more focus including x-axis, y-axis or anywhere in the figure. The size of the asterisks are not of same size (Example Figure 2 (j), Figure 3 (a-f) etc). Please make it consistent. This issue has been brought into attention in the previous comments as well. 

3) Line 35: See "30cm" correct those spacing error throughout the manuscript.

Overall, I would still suggest the authors to check every minute details on their own before it goes for the publication.

Author Response

Thank you for your kind and helpful advice. Please refer to our response listed below:

The article has been now improved now. However, there are still some improvements the authors need to make.

For example:

  • Line 72: "was showed": I am not checking the English language, please make as needed correction through out the manuscript. I can suggest to run your manuscript with native English speaker or Professionals.

Reply: Thanks. We have gone through this manuscript again. Several minor errors in English are corrected. Please refer to the words in red in our revision.

  • Please consider consistency in the text within the figures to be more focus including x-axis, y-axis or anywhere in the figure. The size of the asterisks are not of same size (Example Figure 2 (j), Figure 3 (a-f) etc). Please make it consistent. This issue has been brought into attention in the previous comments as well. 

Reply:  Thanks for your comment on this. As you may have noticed that different panels in one single figure are different in size, therefore we think it is better not to use the identical font size. However, we tried our best to unify the font size across figures.

3) Line 35: See "30cm" correct those spacing error throughout the manuscript. Overall, I would still suggest the authors to check every minute details on their own before it goes for the publication.

Reply: I guess you meant “30cm” in line 275. We corrected that and other spacing errors. Thanks!
